

# A CT-based integrated model for preoperative prediction of occult lymph node metastasis in early tongue cancer

Wei Han[1,2,*], Yingshu Wang[3,*], Tao Li[2], Yuke Dong[2], Yanwei Dang[2], Liang He[1], Lianfang Xu[2], Yuhao Zhou[2], Yujie Li[2] and Xudong Wang[1]

[1] Department of Maxillofacial and Otorhinolaryngological Oncology, Tianjin Medical University Cancer Institute and Hospital, National Clinical Research Center for Cancer, Tianjin Clinical Research Center for Cancer, Key Laboratory of Cancer Prevention and Therapy, Tianjin, China
[2] Department of Otolaryngology, Head and Neck Surgery, Zhengzhou Central Hospital Affiliated to Zhengzhou University, Zhengzhou, China
[3] Department of Radiology, the Affiliated Cancer Hospital of Zhengzhou University, Zhengzhou, China
* These authors contributed equally to this work.

Corresponding author
Xudong Wang, wxd.1133@163.com

## ABSTRACT

**Background:** Occult lymph node metastasis (OLNM) is an essential prognostic factor for early-stage tongue cancer (cT1-2N0M0) and a determinant of treatment decisions. Therefore, accurate prediction of OLNM can significantly impact the clinical management and outcomes of patients with tongue cancer. The aim of this study was to develop and validate a multiomics-based model to predict OLNM in patients with early-stage tongue cancer.

**Methods:** The data of 125 patients diagnosed with early-stage tongue cancer (cT1-2N0M0) who underwent primary surgical treatment and elective neck dissection were retrospectively analyzed. A total of 100 patients were randomly assigned to the training set and 25 to the test set. The preoperative contrast-enhanced computed tomography (CT) and clinical data on these patients were collected. Radiomics features were extracted from the primary tumor as the region of interest (ROI) on CT images, and correlation analysis and the least absolute shrinkage and selection operator (LASSO) method were used to identify the most relevant features. A support vector machine (SVM) classifier was constructed and compared with other machine learning algorithms. With the same method, a clinical model was built and the peri-tumoral and intra-tumoral images were selected as the input for the deep learning model. The stacking ensemble technique was used to combine the multiple models. The predictive performance of the integrated model was evaluated for accuracy, sensitivity, specificity, and the area under the receiver operating characteristic curve (AUC-ROC), and compared with expert assessment. Internal validation was performed using a stratified five-fold cross-validation approach.

**Results:** Of the 125 patients, 41 (32.8%) showed OLNM on postoperative pathological examination. The integrated model achieved higher predictive performance compared with the individual models, with an accuracy of 84%, a sensitivity of 100%, a specificity of 76.5%, and an AUC-ROC of 0.949 (95% CI [0.870–1.000]). In addition, the performance of the integrated model surpassed that of younger doctors and was comparable to the evaluation of experienced doctors.

**Conclusions:** The multiomics-based model can accurately predict OLNM in patients with early-stage tongue cancer, and may serve as a valuable decision-making tool to determine the appropriate treatment and avoid unnecessary neck surgery in patients without OLNM.

## INTRODUCTION

Tongue cancer affects the oral and maxillofacial area and is more common in men due to the higher frequency of tobacco and alcohol consumption. Treatment for early-stage tongue cancer usually involves surgery, radiation therapy, or a combination of both (*Coletta, Yeudall & Salo, 2020*; *Ansarin et al., 2019*; *Pfister et al., 2020*). Cervical lymph node metastasis is a major concern in the early stage of tongue cancer and can reduce patient survival by almost half. It is currently the primary prognostic factor for tongue cancer (*Ren, Yuan & Tao, 2022*; *Imai et al., 2017*; *Li et al., 2023*). Occult lymph node metastasis (OLNM) is diagnosed with the presence of micrometastasis in the lymph nodes through pathological examination after neck dissection, before which the results of palpation, ultrasound, CT, MRI, and other imaging tests are negative (*Baba et al., 2020*). About 40–60% of the patients in the early stage of oral cancer have OLNM (*Yuan, Ren & Tao, 2021*; *Ren, Yuan & Tao, 2022*; *Doll et al., 2022*). In addition, patients without pathological lymph node metastasis (pN0) who undergo neck dissection have to bear unnecessary intraoperative risks and postoperative complications such as shoulder dysfunction. On the other hand, a lack of effective treatment in patients without clinical lymph node metastasis (cN0) but with OLNM may lead to cervical lymph node metastasis or even extracapsular spread during follow-up, resulting in a worse prognosis (*Wang et al., 2022*). Therefore, early detection and accurate prediction of OLNM are critical for determining the optimal treatment and improving patient outcomes.

In recent years, radiomics and deep learning have helped identify early signs of metastasis and predict the likelihood of cancer recurrence (*Ferreira-Junior et al., 2020*, *Jalalifar et al., 2022*; *Bang et al., 2023*). Radiomics involves the extraction of quantitative features from medical images to identify subtle patterns that are not visible to the naked eye (*Guiot et al., 2022*; *Currie & Rohren, 2022*; *Currie, Hawk & Rohren, 2023*). Deep learning refers to training deep neural networks, such as conventional convolutional neural network (CNN) on large datasets, and fine-tuning them on smaller datasets to improve the predictive ability (*Hosny et al., 2018*; *Liu et al., 2020*). It uses an optimization algorithm called back-propagation to adjust its internal parameters for the best image output prediction.

Combining radiomics with deep learning can help improve the accuracy of the prediction models for lymph node metastasis in early tongue cancer (*Gillies, Kinahan & Hricak, 2016*; *Leger et al., 2017*). Machine learning algorithms can identify patterns
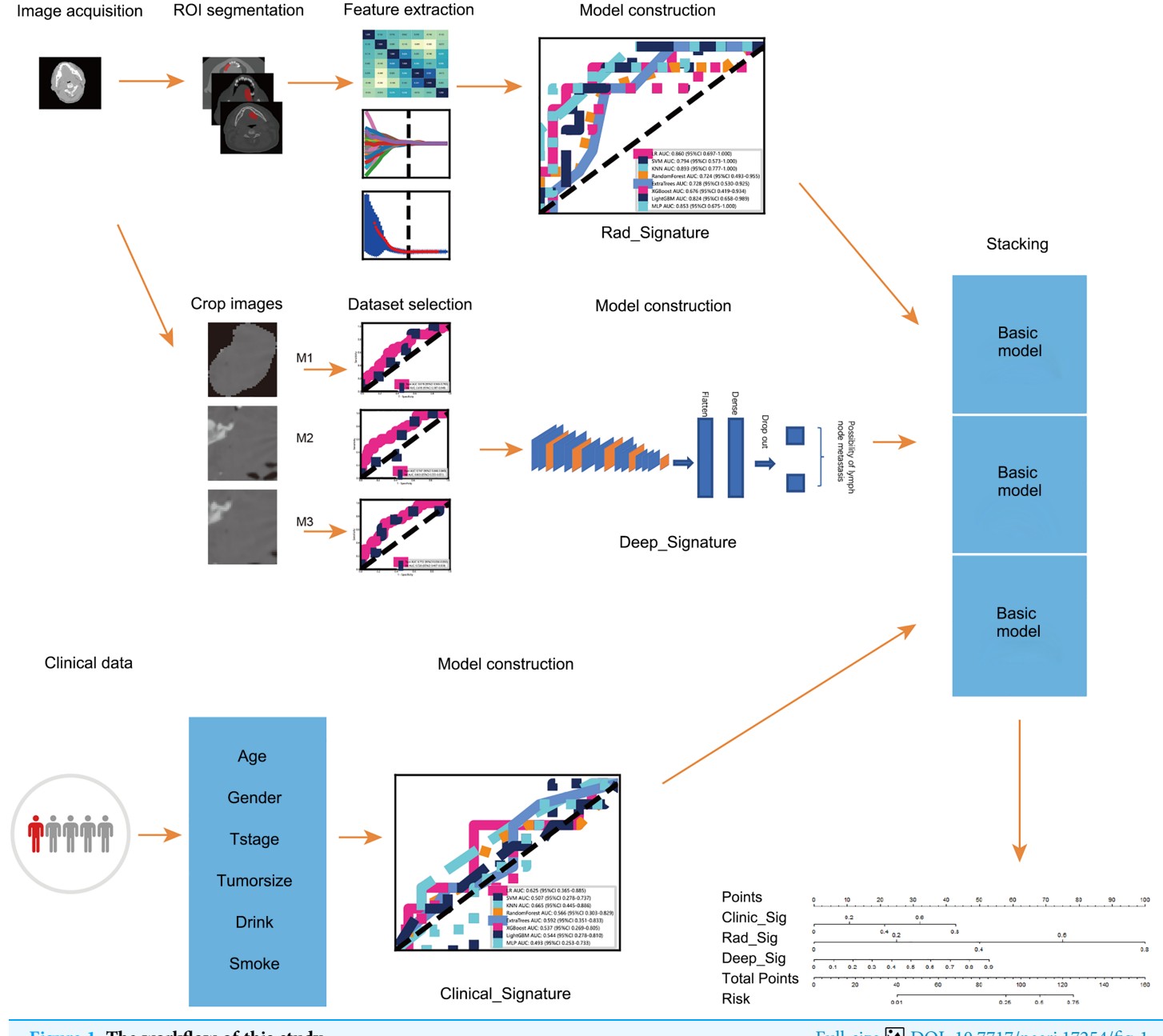

**Figure 1 The workflow of this study.**

indicative of metastasis by analyzing radiomics features, such as texture, shape, and intensity (*Tang et al., 2022*). Furthermore, deep transfer learning can automatically extract more meaningful features from medical images to avoid massive engineering of radiomics for more accurate predictions in a shorter duration, achieving good performance (*Hosny et al., 2018*).

Previous studies have used this approach on the intratumoral region without considering the peritumoral area. However, there is evidence that peritumoral features can improve diagnostic efficiency (*Zhou et al., 2018*; *Zhang et al., 2022*; *Zhao et al., 2023*).

*Zhang et al. (2022)* demonstrated that time-serial CT-based radiomics signature integrating intra- and peri-tumoral features offered the potential to predict progression-free survival for lung adenocarcinoma patients treated with epidermal growth factor receptor-tyrosine kinase inhibitors. *Zhao et al. (2023)* developed and validated a nomogram based on intratumoral and peritumoral radiomics signatures to predict distant metastasis-free survival in patients with locally advanced rectal cancer before the treatment with neoadjuvant chemoradiotherapy. In addition, *Zhou et al. (2018)* generated a radiogenomics map for non-small cell lung cancer to illustrate the relationship between molecular and imaging phenotypes.

This study aimed to combine radiomics and deep transfer learning to predict OLNM in patients with early-stage tongue cancer based on contrast-enhanced CT images. The peritumoral region was incorporated into the radiomics-based model to improve the predictive accuracy. In clinical settings, this model can minimize decision-making errors by surgeons and reduce the risk of unnecessary surgery. The workflow of the study is shown in Fig. 1.

## MATERIALS AND METHODS

### Participants

A retrospective study was conducted using data on patients diagnosed with tongue cancer between 2015 and 2022 at the Tianjin Medical University Cancer Institute and Hospital and Zhengzhou Central Hospital Affiliated to Zhengzhou University. A total of 125 patients were randomly divided into the training set (100 cases) and the test set (25 cases). The study was approved by the ethics committee of Zhengzhou Central Hospital (202336) and conducted in line with the Declaration of Helsinki. Due to the retrospective nature of the study, the need for informed consent was waived by the committee.

The inclusion criteria were as follows: (1) The participants must have a pathological diagnosis of tongue squamous cell carcinoma, (2) and were treated with planned surgery and elective neck dissection, (3) with a clinical diagnosis of cT1-2N0M0 based on palpation and imaging tests. (4) The contrast-enhanced CT scans of the head and neck area were performed within 30 days prior to surgery, and (5) the complete clinical information was available. Patients with a history of previous malignancy, or with unidentifiable lesions or artifacts-induced radiological interference on CT scans were excluded. The CT pictures from hospital were collected using a GE Medical Systems Light Speed Pro 12 CT scanner with 120 kVp and 150 mAs. The parameters for the CT scan were as follows: scanning slice thickness 1.5 mm, pitch of 1.5 mm, and matrix of 512 × 512. The baseline characteristics including gender, sex, T stage, tumor size, smoking history, and alcohol consumption were retrieved from the databases of the hospitals.

### Region of interest (ROI) segmentation and preprocessing

Two radiologists blinded to the clinicopathological information of the patients independently conducted tumor segmentation on the CT dataset using ITK-SNAP version 3.8.0 (http://www.itksnap.org). A senior radiologist verified all delineation tags and corrected any errors. Since the images came from two CT scanners with different

parameters, resampling was done before the features could be extracted from the ROI of the CT. The image space was normalized to $1 \times 1 \times 1$. The inter-observer reliability and intra-observer repeatability of the extraction of radiomic characteristics were generally evaluated using inter- and intra-class correlation coefficients (ICCs). Features with ICC > 0.75 were retained for good consistency. The extracted image features were then normalized by adjusting the distribution of all features to the same range. The data were standardized according to the formula: $Z = (X-\mu)/\sigma$, where $\mu$ is the feature's mean and $\sigma$ is the feature's standard deviation.

## Feature extraction and selection

The radiomics features were classified based on geometry, intensity, and texture to describe different aspects of the tumors, specifically from the following aspects: (I) geometry—a three-dimensional form of the tumor; (II) intensity—first-order statistical distribution of the voxel intensities within the tumor; (III) texture—the patterns, or second- and high-order spatial distributions of the intensities. The neighborhood gray-tone difference matrix, gray-level co-occurrence matrix, gray-level run length matrix, gray level size zone matrix, and other techniques were used to extract the texture features (Fig. 2A).

The radiomics features with $p$ values less than 0.05 in the Mann-Whitney U test were screened. The correlation between these features was analyzed by Spearman's method, and those with a correlation coefficient >0.9 were selected. A greedy recursive deletion strategy was used for feature filtering, and the feature with the highest level of duplication in the current set was removed each time to preserve the ability to display features as fully as possible. Finally, 115 features were retained and subjected to least absolute shrinkage and selection operator (LASSO) regression with the scikit-learn package in Python using 10-fold cross validation with minimum criteria. The features with non-zero coefficients in the regression model were used to construct the radiomics signature, and a linear combination of retained features weighted by their model coefficients was used to calculate the radiomics score for each patient (Figs. 2B–2D).

To construct deep transfer learning models, the most representative images were identified. The images were processed through three steps: M1: Assuming that the image background would create interference, the background pixels were removed and the primary lesion area was extracted along with the area surrounding the tumor. M2: The largest ROI was selected from the CT images and enlarged by five pixels. M3: In the 3D image, the tumor area was expanded evenly by two voxels to obtain the peritumoral region, and the image with the largest ROI was selected (Fig. 2E).

## Development of different models

Radiomics model: The final features were input into various machine learning models such as support vector machine (SVM), k-nearest neighbor (kNN), random forest, extra trees, extreme gradient boosting (XGBoost), light gradient boosting machine (LightGBM), multilayer perceptron (MLP), and logistic regression (LR) to construct the radiomics model (*Cortes & Vapnik, 1995*; *Geurts, Ernst & Wehenkel, 2006*; *Chen & Guestrin, 2016*; *Ke et al., 2017*).
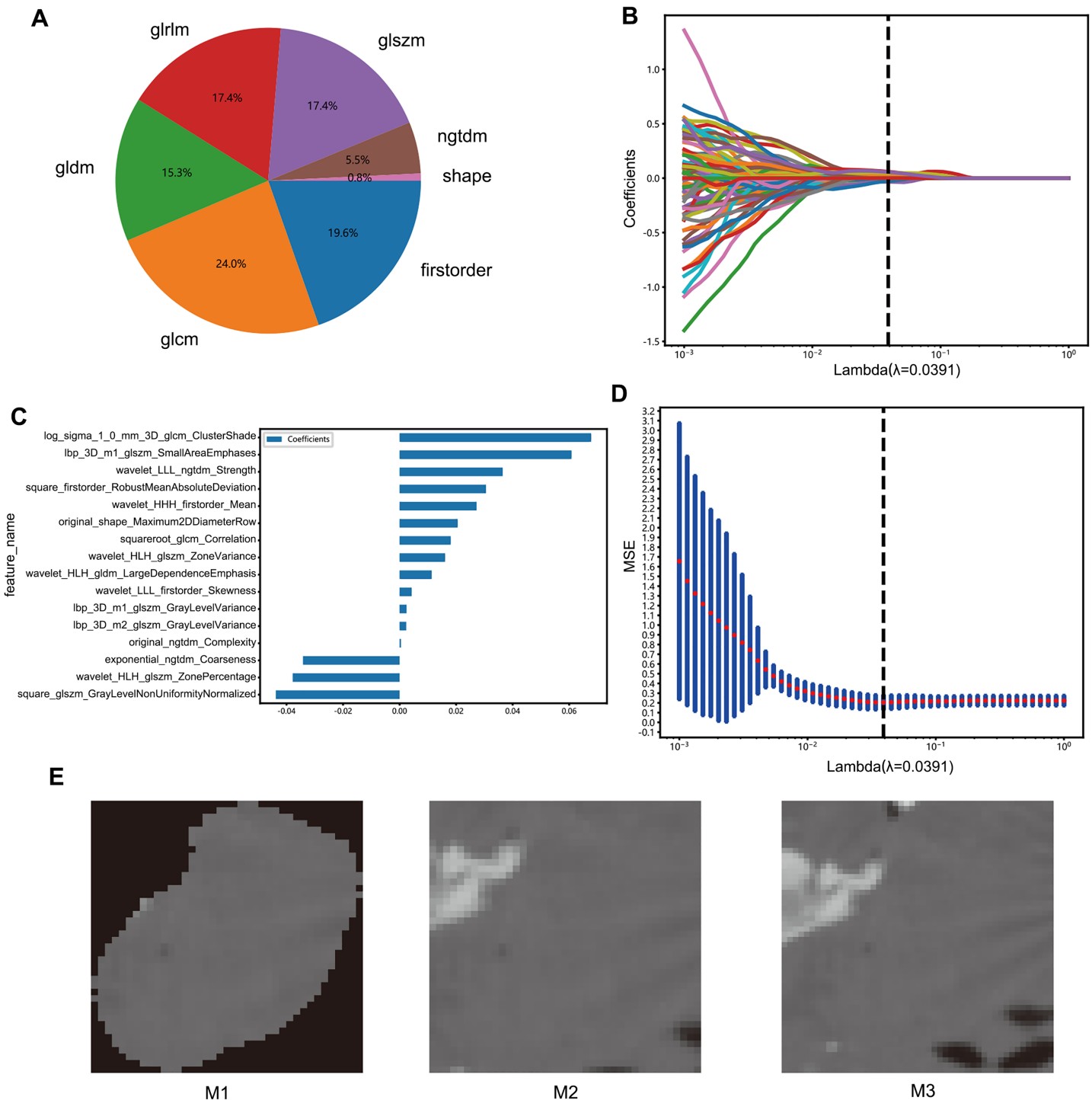

**Figure 2 Features selection process.** (A) Number and ratio of handcrafted features; nonzero coefficients were selected to establish the Rad-score with a LASSO logistic regression model. (B) Coefficients and (D) MSE (mean standard error) of 10 folds validation. (C) The histogram of the Rad-score based on the selected features. (E) Deep learning datasets based on different methods (M1, M2, M3).

Clinical model: The preoperative clinical data included age, gender, T stage, tumor size, drinking and smoking history. The same methodology described above was used to construct the clinical model.

Deep transfer learning model: Based on literature review and preliminary experimental testing, ResNet50 (*He et al., 2016*) was selected as the deep transfer learning model, and three datasets were incorporated for training. The stochastic gradient descent (SGD) optimizer was adopted with a learning rate of 0.001 and 100 epochs. A fully connected layer, followed by a dropout layer with a 0.5 drop rate, was employed to prevent overfitting. The optimal dataset was selected for training, and the structured data were extracted from the average pooling layer (*i.e.*, the penultimate layer). The deep learning model was then constructed using the same methodology mentioned above. Finally, the three base models were integrated using the stacking strategy to obtain the integrated model.

## Statistical analysis

Student's t-test or the Mann-Whitney U test was used to compare continuous variables, and the chi-squared test was used for categorical variables. All statistical tests were two-tailed and $P < 0.05$ was considered statistically significant. Python 3.7 was used for statistical analysis.

# RESULTS

## Clinical data

A total of 125 patients met the inclusion criteria, of which 100 were randomly assigned to the training set and 25 to the test set. The clinical features of the patients are summarized in Table 1.

## Performance of the radiomics model

The LightGBM model demonstrated the best performance and was selected to construct the radiomics signature, with the area under curve (AUC) values of 0.896 (95% CI [0.825–0.967]) and 0.824 (95% CI [0.658–0.989]) in the training and testing datasets, respectively (Fig. 3A). The confusion matrix for LightGBM is shown in Fig. 3B. The AUC of the clinical signature was 0.728.

## Performance of the deep learning model

In the deep learning models, the performance of M3 was significantly higher than that of other datasets, with an AUC value of 0.826 in the training cohort and 0.674 in the test cohort (Fig. 4). Subsequently, the deep transfer learning model trained on the M3 dataset was selected as the deep signature.

## The integrated model performed better than a single model

Once the base models were trained, we used stacking to combine their predictions (*Džeroski & Ženko, 2004*). The dataset was split into the training and test sets, and the predictions generated from each base model on the test set were used as input features to

**Table 1** The detailed clinical characteristics of the patients.

|  | train-label = ALL | train-label = 0 | train-label = 1 | p value | test-label = ALL | test-label = 0 | test-label = 1 | p value |
|---|---|---|---|---|---|---|---|---|
| Age | 58.54 ± 12.65 | 59.61 ± 10.54 | 56.36 ± 16.09 | 0.229041 | 54.92 ± 15.39 | 56.18 ± 16.05 | 52.25 ± 14.54 | 0.563049 |
| Tumor size | 2.10 ± 0.91 | 2.03 ± 1.01 | 2.24 ± 0.68 | 0.286474 | 1.99 ± 0.73 | 1.74 ± 0.68 | 2.53 ± 0.53 | 0.008355 |
| Gender |  |  |  | 0.876726 |  |  |  | 0.479802 |
| Male | 58 (58.00) | 38 (56.72) | 20 (60.61) |  | 18 (72.00) | 11 (64.71) | 7 (87.50) |  |
| Female | 42 (42.00) | 29 (43.28) | 13 (39.39) |  | 7 (28.00) | 6 (35.29) | 1 (12.50) |  |
| T stage |  |  |  | 0.056452 |  |  |  | 0.479802 |
| 1 | 29 (29.00) | 24 (35.82) | 5 (15.15) |  | 7 (28.00) | 6 (35.29) | 1 (12.50) |  |
| 2 | 71 (71.00) | 43 (64.18) | 28 (84.85) |  | 18 (72.00) | 11 (64.71) | 7 (87.50) |  |
| Drink |  |  |  | 0.859987 |  |  |  | 1 |
| No | 67 (67.00) | 44 (65.67) | 23 (69.70) |  | 14 (56.00) | 10 (58.82) | 4 (50.00) |  |
| Yes | 33 (33.00) | 23 (34.33) | 10(30.30) |  | 11 (44.00) | 7 (41.18) | 4 (50.00) |  |
| Smoke |  |  |  | 0.993163 |  |  |  | 0.378313 |
| No | 56 (56.00) | 37 (55.22) | 19 (57.58) |  | 11 (44.00) | 9 (52.94) | 2 (25.00) |  |
| Yes | 44 (44.00) | 30 (44.78) | 14 (42.42) |  | 14 (56.00) | 8 (47.06) | 6 (75.00) |  |

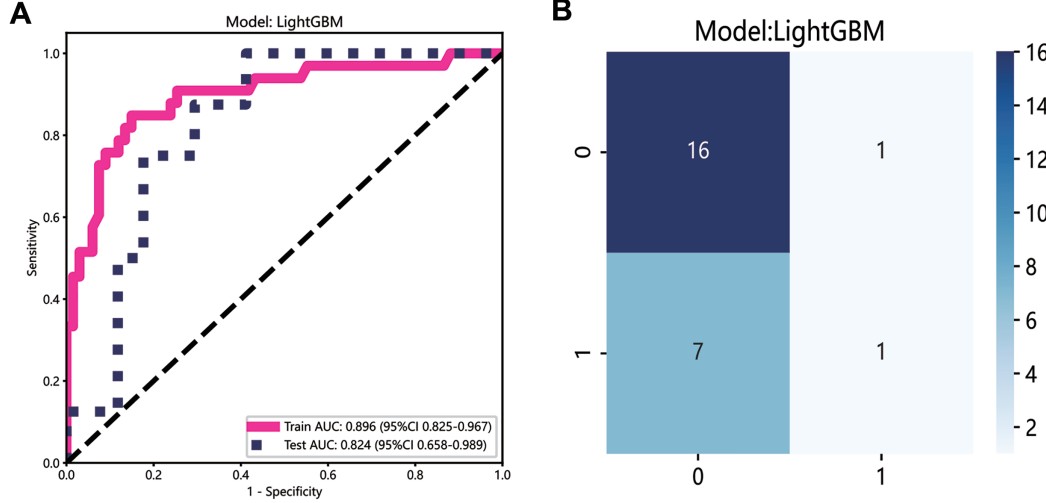

**Figure 3** **Development of radiomics model.** (A) The ROC curves of the radiomics model; (B) the confusion matrix of predicting lymph node metastasis.

train a meta-model for optimally combining the predictions of the three base models. Logistic regression was used as the meta-model for its simplicity and interpretability. Our integrated model demonstrated improved performance compared with individual models, achieving an accuracy of 95%, a sensitivity of 100%, and a specificity of 92.5% in the training dataset, and 84% accuracy, 100% sensitivity, and 76.5% specificity in the test dataset (Table 2). The AUC of the integrated model was 0.949 (95% CI: [0.870–1.000]), which was higher than the AUCs of the radiomics (0.893), clinical (0.728), and deep learning models (0.798) (Figs. 5A and 5B).

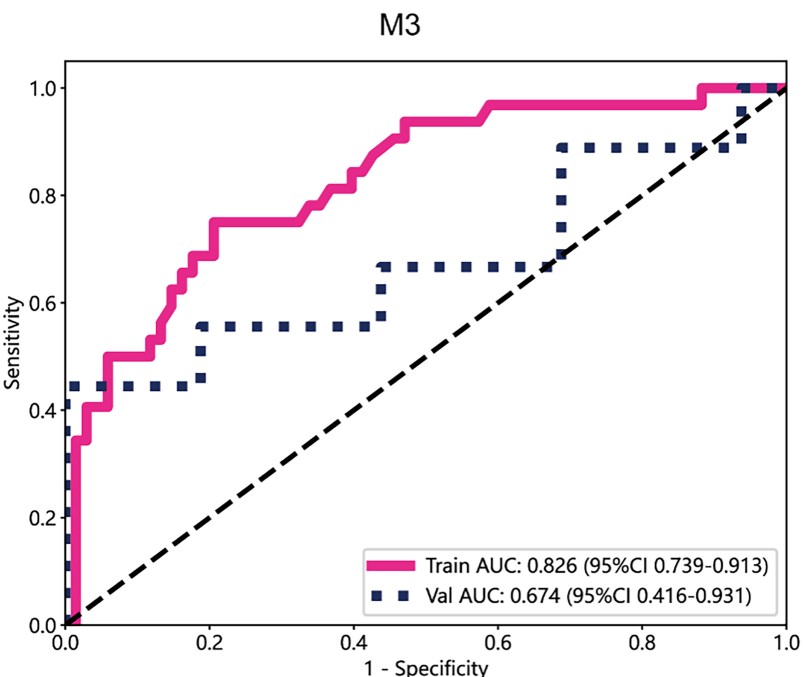

**Figure 4  The ROC curves of M3.**

**Table 2  The performance of different models.**

|  | Accuracy | AUC | 95% CI | Sensitivity | Specificity | Recall | F1 | Cohort |
|---|---|---|---|---|---|---|---|---|
| Clinic sig | 0.720 | 0.815 | [0.725–0.904] | 0.848 | 0.657 | 0.848 | 0.667 | Train |
| Rad sig | 0.850 | 0.896 | [0.825–0.967] | 0.848 | 0.851 | 0.848 | 0.789 | Train |
| Deep sig | 0.930 | 0.980 | [0.960–1.000] | 0.970 | 0.910 | 0.970 | 0.901 | Train |
| Nomogram | 0.950 | 0.989 | [0.976–1.000] | 1.000 | 0.925 | 1.000 | 0.930 | Train |
| Clinic sig | 0.680 | 0.728 | [0.526–0.930] | 0.875 | 0.588 | 0.875 | 0.636 | Test |
| Rad sig | 0.720 | 0.893 | [0.777–1.000] | 1.000 | 0.588 | 1.000 | 0.696 | Test |
| Deep sig | 0.720 | 0.798 | [0.625–0.971] | 1.000 | 0.625 | 1.000 | 0.696 | Test |
| Nomogram | 0.840 | 0.949 | [0.870–1.000] | 1.000 | 0.765 | 1.000 | 0.800 | Test |

## Compared with a single model, the integrated model showed superior discriminative ability and produced larger net benefits

The performance of the integrated model was also evaluated by the DeLong test and decision curve analysis (DCA). The DeLong test indicated that the integrated model had a higher AUC compared with the individual models (nomogram *vs* clinical: 0.002; nomogram *vs* radiomics: 0.073; nomogram *vs* deep learning: 0.035), demonstrating the superior discriminative ability of the integrated model. The DCA further showed that our integrated model yielded a greater net benefit compared with other models across the majority of practical threshold probabilities (Fig. 6). The nomogram of the integrated model is shown in Fig. 7.
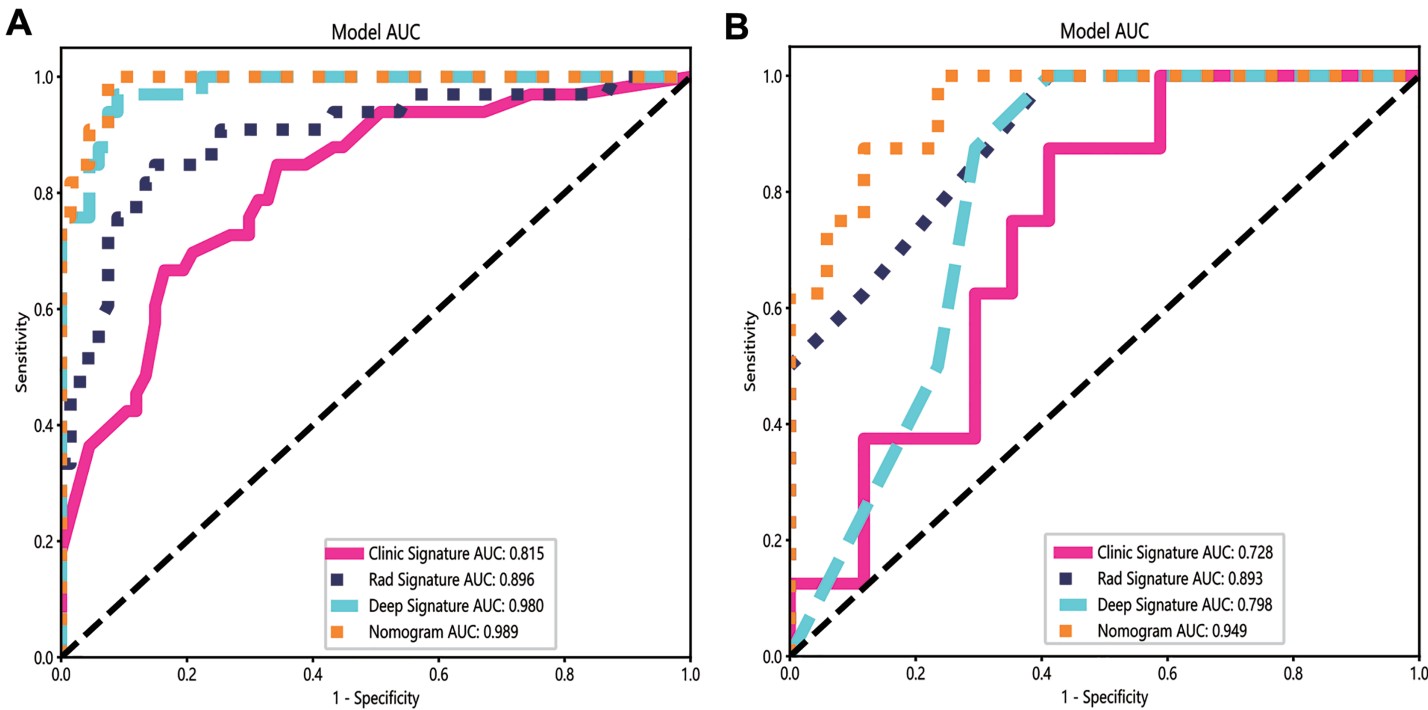

**Figure 5** **The ROC curves of the clinical model, radiomics model, deep learning model and combined model in the training cohort (A) and the validation cohort (B).** The integrated model demonstrated significantly higher AUCs in the training and validation cohorts (AUCs of 0.989 and 0.949) than the clinical model (AUCs of 0.815 and 0.728), the radiomics model (AUCs of 0.896 and 0.893) and the deep learning model (AUCs of 0.980 and 0.798).

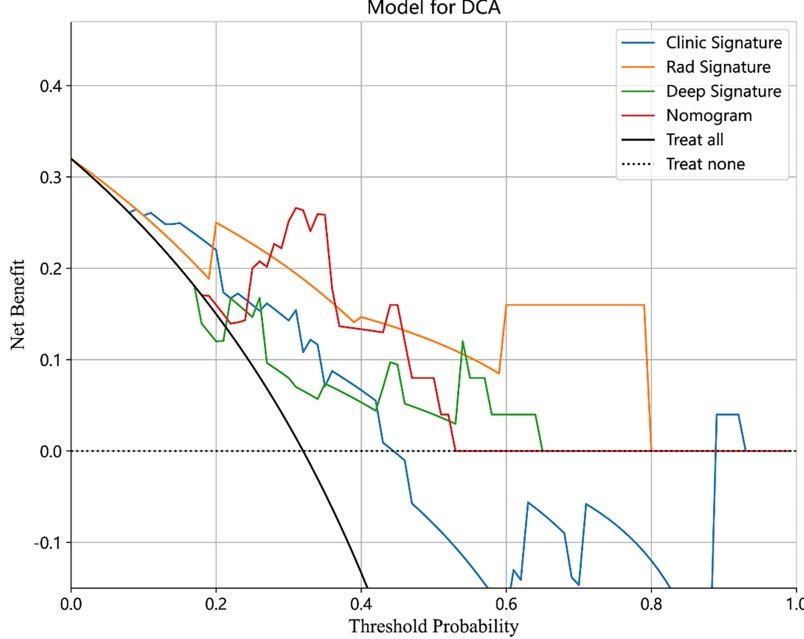

**Figure 6** **The DCA of the clinical model, radiomics model, deep learning model and combined model.**

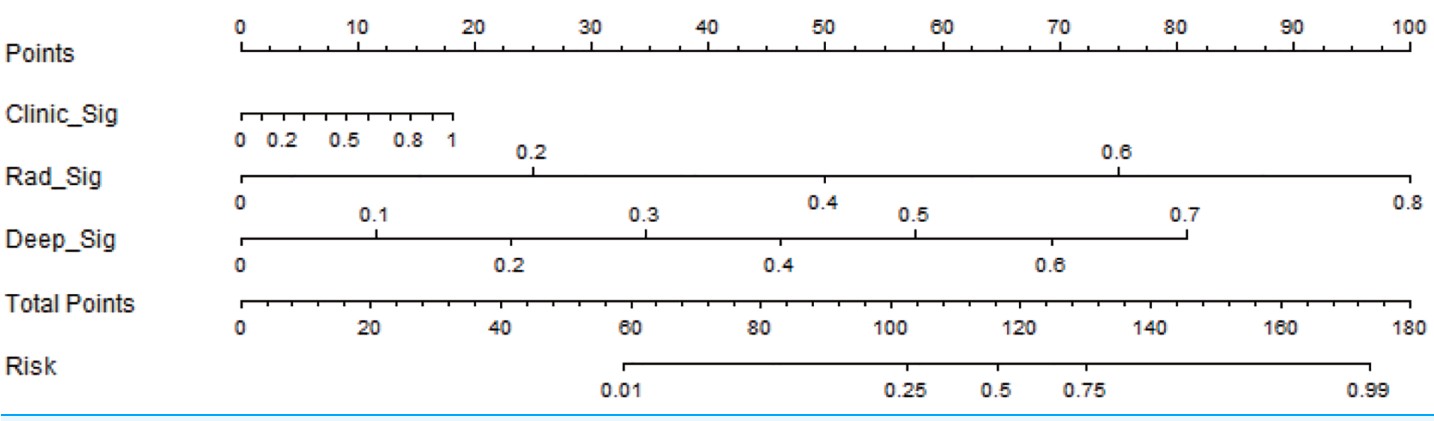

**Figure 7 The development of the nomogram for clinical use.**

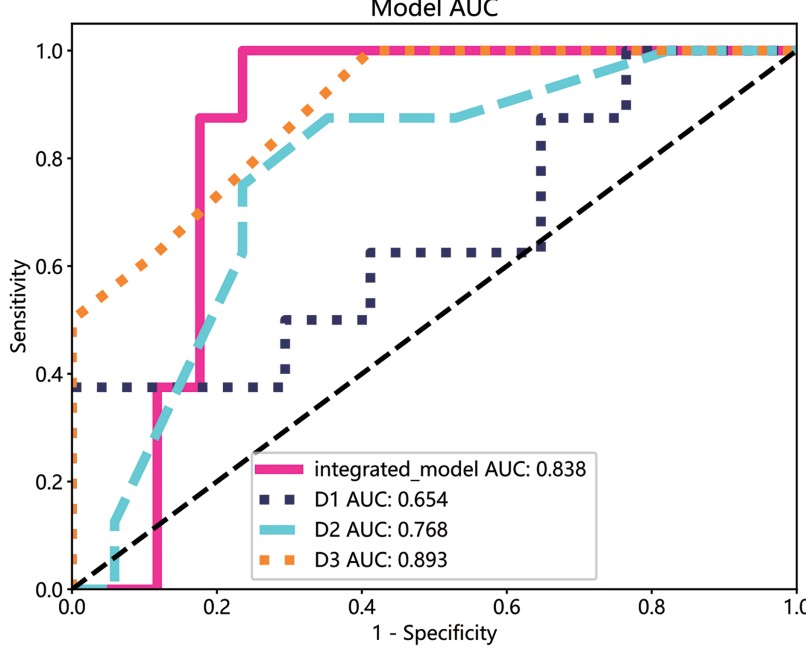

**Figure 8 The ROC curves of the integrated model and diagnostic results of each radiologist in the test cohort.** The integrated model reached a high AUCs (AUC: 0.838), doctor1 acquired 0.654 of AUC, the AUC of doctor2 was 0.768 and doctor3 demonstrated the highest with 0.893.

## Comparison of the integrated model with clinical assessment

The performance of the integrated model was compared with that of clinical experts. Three clinicians with varying levels of experience in oral diseases, *i.e.*, less than 5 years (D1), less than 10 years but longer than D1 (D2), and more than 15 years (D3), were asked to interpret the imaging and clinical data. The decision of the three clinicians was compared with that of the integrated model by plotting ROC curves. As shown in Fig. 8, the integrated model achieved an AUC of 0.838 compared with 0.654, 0.768, and 0.893 for clinicians D1, D2, and D3, respectively, suggesting differences between the model and the expert evaluation.

## DISCUSSION

OLNM is an essential prognostic factor in early-stage tongue cancer (cT1-2N0M0) and influences treatment decisions (*Kubo et al., 2022*, *Kwon, 2022*). Thus, accurate prediction of OLNM can significantly impact clinical management and outcomes of tongue cancer patients (*Baba et al., 2019*). In this study, we developed a decision-level fusion system for classifying CT images by integrating radiomics, clinical, and deep learning models with the stacking ensemble technique. A nomogram was employed to visualize the integrated prediction results and allow clinical interpretation. To our knowledge, this is also the first time that an integrated model capturing the features of CT images around the tumor has been used to predict OLNM in early-stage tongue cancer.

The integrated model exhibited superior performance compared with the individual radiomics, clinical, and deep learning models in terms of the AUC values, and this can be attributed to the combination of the complementary strengths of the three models that allowed diverse aspects of the data to be captured. For instance, the radiomics model allowed quantitative features to be extracted from CT images and subtle patterns that might be overlooked by human observers to be captured, while the clinical model provided additional demographic and clinical information. We also applied deep learning to predict the presence of OLNM in cT1-2N0 patients (*Jin et al., 2021*). The deep learning model based on ResNet50 can directly analyze raw medical images and learn complex and high-level representations of the data (*Yu et al., 2020*). However, the format of the raw images is a matter of debate. While some researchers advocate for directly incorporating 3D images into the model, others believe that the images should first undergo a flattening process before being fed into the model (*Harmon et al., 2020*; *Calabrese et al., 2022*; *Tian et al., 2020*; *Shi et al., 2022*; *Cho et al., 2022*; *Charles et al., 2017*). However, direct use of 3D data for deep learning training often requires significant computational resources, such as high-performance graphic processing units (GPUs) and large memory capacities, making the training and deployment processes expensive and time-consuming. Furthermore, medical image datasets are often small and imbalanced, possibly leading to overfitting when 3D data are directly used for deep learning training. Overfitting can cause the model to perform well with the training data but underperform in real-world applications due to reduced generalization. Therefore, we converted the 3D CT images to 2D images, during which determination of the feature extraction range was required due to relatively fewer features. To extract more representative features, we adopted three cropping methods to obtain suitable images as the input for the deep learning model. The M1 dataset was first obtained by setting pixel values of the background to zero. Then, the largest cross-section from the 3D ROI was selected and expanded by five pixels to obtain dataset M2. Finally, the 3D ROI was evenly expanded by two voxels around the tumor, and the largest cross-section was cropped to generate the M3 dataset. Incorporating peri-tumoral information can prevent overfitting when deep learning models are applied to most small medical imaging datasets by increasing the number of features. Among the two peri-tumoral processing methods, M3 uniformly expanded a fixed length in 3D

images, while M2 involved rectangular expansion in 2D images. Consequently, M3 encompassed more critical features than M2, eventually yielding superior results.

By combining the outputs of multiple base models, stacking leverages the strengths of each to achieve better predictive accuracy (*Mostafaei et al., 2020*) and reduces overfitting by diversifying the model ensemble, leading to more robust and generalizable results. We compared the performance of our integrated model with the decision-making abilities of clinicians and found that the integrated model had superior accuracy. As already explained, this can be attributed to the ability of machine learning algorithms to detect subtle patterns and relationships in the data that may not be easily discernible by human experts. Our findings indicate that a targeted multiomics integration model may help clinical practitioners, especially those with less experience, attain a performance level similar to that of experts. Clinicians often need to analyze vast amounts of information, including medical images, clinical records, and laboratory results, within a limited time, which can lead to cognitive overload and increase the likelihood of errors. The integrated model can rapidly process and analyze a large number of data, thus reducing the time needed for diagnosis and treatment planning. Another benefit of the integrated model is its ability to continuously learn and adapt new data that become available. As medical knowledge and clinical practice evolve, clinicians need to update their knowledge and decision-making process accordingly. However, this process can be time-consuming and challenging. The integrated model can be retrained with new data, enabling it to stay up-to-date with the latest advancements in the field and provide more accurate predictions. While the expertise of clinicians is invaluable, personal biases and preferences can sometimes influence their decisions, potentially leading to sub-optimal patient outcomes. By providing an objective, data-driven approach, the integrated model can minimize the impact of subjectivity in decision-making, leading to more consistent and optimal patient care. The nomogram incorporated the predictions from the radiomics, clinical, and deep learning models, as well as the output from the meta-model, to provide a comprehensive and intuitive representation of the decision-level fusion process. This can enable more effective communication of the results to both clinical practitioners and researchers, facilitating better understanding and application of the model in real-world settings.

In clinical practice, clinical data and CT image features of patients with early tongue cancer can be collected before surgery. Through objective data analysis and processing, the integrated model can be used to predict OLNM more accurately in a short period, and unnecessary neck surgery can be minimized for patients with negative OLNM. During the follow-up period, new data can be continuously collected for prediction, patients with missed diagnosis or OLNM are detected as early as possible, and lymph node dissection can be completed in time to effectively improve the prognosis. For patients with positive OLNM, cervical surgery should be completed simultaneously to reduce the risk of cervical lymph node metastasis and spread during follow-up. The integrated model is beneficial to the standardized and individualized diagnosis and treatment of early tongue cancer and improves the curative effect and prognosis.

However, several limitations to our study should be acknowledged. First, the sample size used for training and validation was relatively small, and this may limit the generalizability

of our findings. Future studies should include larger, more diverse patient cohorts and include predictive molecular biomarkers associated with tissue or blood to further optimize the synthesis model, and prospective studies are needed to verify its efficacy. Second, the choice of the meta-model in the stacking ensemble technique could potentially impact the results. In our study, we employed logistic regression as the meta-model due to its simplicity and interpretability. However, other meta-models, such as SVM or random forest, could be explored in the future to assess their impact on the performance of the integrated model.

## CONCLUSION

The multiomics-based model demonstrated high accuracy, sensitivity, and specificity for predicting OLNM in patients with early-stage tongue cancer. Despite the noted limitations, our integrated model could serve as a valuable tool for improving diagnosis and treatment planning, eventually leading to better patient outcomes.

### Funding
The authors received no funding for this work.

### Competing Interests
The authors declare that they have no competing interests.

### Author Contributions
- Wei Han conceived and designed the experiments, performed the experiments, analyzed the data, prepared figures and/or tables, authored or reviewed drafts of the article, and approved the final draft.
- Yingshu Wang conceived and designed the experiments, performed the experiments, analyzed the data, prepared figures and/or tables, authored or reviewed drafts of the article, and approved the final draft.
- Tao Li performed the experiments, analyzed the data, prepared figures and/or tables, and approved the final draft.
- Yuke Dong performed the experiments, analyzed the data, prepared figures and/or tables, and approved the final draft.
- Yanwei Dang performed the experiments, analyzed the data, prepared figures and/or tables, and approved the final draft.
- Liang He performed the experiments, prepared figures and/or tables, and approved the final draft.
- Lianfang Xu performed the experiments, prepared figures and/or tables, and approved the final draft.
- Yuhao Zhou performed the experiments, prepared figures and/or tables, and approved the final draft.
- Yujie Li performed the experiments, analyzed the data, prepared figures and/or tables, and approved the final draft.

# PeerJ

- Xudong Wang conceived and designed the experiments, authored or reviewed drafts of the article, and approved the final draft.

## Human Ethics

The following information was supplied relating to ethical approvals (*i.e.*, approving body and any reference numbers):

The study was approved by the ethics committee of Zhengzhou Central Hospital (202336).

## Data Availability

The raw measurements are available in the Supplemental File.

## Supplemental Information

Supplemental information for this article can be found online at http://dx.doi.org/10.7717/peerj.17254#supplemental-information.

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
