# Peer review of "A CT-based integrated model for preoperative prediction of occult lymph node metastasis in early tongue cancer"

_PeerJ, doi:10.7717/peerj.17254_

## Round 0.1 · original submission · Major Revisions

Some important comments have been provided by the reviewers, and should be carefully addressed.

**Language Note:** The review process has identified that the English language must be improved. PeerJ can provide language editing services - please contact us at copyediting@peerj.com for pricing (be sure to provide your manuscript number and title). Alternatively, you should make your own arrangements to improve the language quality and provide details in your response letter. – PeerJ Staff

Reviewer 1 ·

Basic reporting

The article introduces a CT-based integrated model for predicting occult lymph node metastasis in early tongue cancer. It stands out for its innovative approach in medical imaging, potentially revolutionizing how early-stage tongue cancer is diagnosed and treated. The research is very meaningful, and there are still some problems that need to be solved.

1. The manuscript effectively introduces a CT-based integrated model for predicting occult lymph node metastasis in early tongue cancer, which is commendable for its clinical relevance. However, it is noticed that various abbreviations are used throughout the document. For clarity and accessibility, especially for readers not familiar with specific medical or technical terms, it is recommended that all abbreviations be spelled out in their full form upon their first occurrence. This includes abbreviations like OLNM, SVM, and others related to the machine learning and medical terms used in the study.

2. The manuscript is comprehensive but at times becomes convoluted due to the use of complex sentences with multiple clauses. Simplifying these sentences will enhance readability and ensure that the material is more approachable for a broader audience, including those who may not have specialized knowledge in this specific area of oncology or radiomics.

3. While the discussion provides valuable insights, particularly about the integrated model's predictive performance and its comparison with expert assessments, it could benefit from further refinement. It would be helpful to more explicitly discuss the implications of these findings for clinical practice, such as how this model could influence treatment decisions or reduce unnecessary surgeries. Also, a discussion on the potential limitations of the study and future research directions would provide a more balanced view.

4. The general quality of English in the manuscript requires improvement. There are instances where the language could be more precise and professional. It is recommended that the manuscript be reviewed and edited by a fluent English speaker with expertise in the subject matter. This would not only enhance the clarity of the text but also ensure that the significance of the research is effectively communicated to an international audience.

5. The manuscript does a good job of addressing ethical considerations and data transparency, especially in terms of human participant checks and data availability. It is suggested to maintain this level of transparency and ethical compliance in future iterations of the manuscript, as it adds credibility and replicability to the research.

Overall, the study presents significant findings with potential clinical implications in the diagnosis and treatment of early-stage tongue cancer. These suggestions aim to enhance the manuscript's impact and accessibility.

Experimental design

Null

Validity of the findings

Null

Additional comments

Null

Reviewer 2 ·

Basic reporting

The article makes a substantial impact in the field of medical research by developing a CT-based integrated model for detecting lymph node metastasis in early-stage tongue cancer. Its primary strengths are its methodological advancement and the potential to significantly enhance diagnostic processes in oncology. The authors present a fascinating ensemble model for predicting occult lymph node metastasis in early-stage tongue cancer. This approach has the potential to revolutionize treatment decisions and reduce unnecessary surgeries. The research is akin to unearthing a hidden treasure in the vast ocean of medical science. However, there are a few areas that could benefit from further refinement:

The 'Results' section, while informative, reads a bit like a mysterious riddle from an ancient tome. A bit more clarity and structure would help in decoding these riddles, making the findings more accessible and understandable to fellow scholars and practitioners. Perhaps, consider adding a 'map' in the form of clear subheadings or bullet points to guide the reader through your treasure trove of results.

While summarizing findings from various studies, your manuscript could transform into an intriguing detective novel, where each piece of evidence is critically analyzed for its quality, biases, and replicability. Picture Sherlock Holmes examining each study with his magnifying glass, providing insightful commentary that enriches the understanding of your model's context in the broader scientific landscape.

The manuscript's linguistic style, though rich in content, could use a touch of refinement. It's as if the manuscript is a rough diamond that needs polishing. Seeking the assistance of a linguistic alchemist, preferably one whose native tongue is English, could transmute the text into a more polished and captivating narrative, enhancing its appeal to a global audience.、

Provide a brief but clear explanation or reference for the statistical methods used, especially for more complex analyses. This will help readers who may not be familiar with these techniques to better understand your methodology.

Dedicate a section to discuss the limitations of your study in more detail. Acknowledging potential weaknesses or constraints will enhance the credibility of your research.

Strengthen your conclusion by succinctly summarizing the key findings and their implications. Avoid introducing new information in this section.

Include a few more recent references to ensure that your literature review is up-to-date. This will demonstrate your engagement with the latest developments in your field.

Adding these elements of refinement and critical analysis, coupled with a linguistic makeover, would elevate your manuscript from being a hidden gem to a shining beacon in the field of medical research.

Experimental design

None

Validity of the findings

None

Additional comments

None

Reviewer 3 ·

Basic reporting

The manuscript is recognized for its significant contribution to oncology, particularly in using a CT-based model to improve prognosis accuracy in early tongue cancer. Its main strength lies in its potential to influence clinical decision-making, offering a new avenue for cancer treatment strategies. The authors applied the multiomics-based model to predict patients with early occult lymph node metastasis tongue cancer more accurately and may serve as a valuable decision-making tool to determine appropriate treatment and avoid unnecessary neck surgery in patients without occult lymph node metastasis. There are some points needed to be addressed as detailed below:

1. The article's use of figures is instrumental in conveying complex data. However, it would be greatly beneficial if the font sizes in these figures were increased for better readability. Additionally, enhancing the overall clarity of these visual representations would align them more closely with the high quality of research presented. On a whimsical note, consider using a font like 'Comic Sans' for a draft version to ensure you pay extra attention to the seriousness of 'Times New Roman' or 'Arial' in the final submission.
2. The document exhibits a strong academic tone and depth. However, minor grammatical errors scattered throughout the manuscript slightly diminish its professional impact. A thorough proofreading, perhaps even by a mythical grammar gnome known for their meticulous attention to detail, would polish the manuscript to meet the high standards of scholarly communication.
3. The discussion section successfully outlines the study's results but would greatly benefit from a more pronounced emphasis on the novelty of your CT-based integrated model. It's akin to finding a new constellation in the sky – the importance of this discovery should be highlighted with clear, starry examples. Similarly, an expanded discussion on the clinical applicability of your findings, perhaps illustrated through hypothetical scenarios or a creative narrative involving time travel to future clinical applications, would significantly enhance the manuscript's practical value.
4. Given the complexity of your ensemble model, a graphical abstract at the beginning of your paper could be incredibly helpful. This should concisely visualize the key aspects of your model and its application in predicting occult lymph node metastasis. Think of it as a quick visual summary that provides a snapshot of your research, aiding readers in grasping the essence of your work at a glance. This addition would be particularly beneficial for busy practitioners who might need to quickly understand the practical implications of your findings.
5. In your results section, the tables present a wealth of data but could benefit from a slight tweak for better readability. Consider increasing the font size in your tables just enough to make them easily readable without having to zoom in. This small adjustment would make the valuable data you've presented more accessible and user-friendly, especially for readers who may not have perfect vision.

Experimental design

none

Validity of the findings

none

---

## Round 0.2 · Minor Revisions

Several essential points have been provided by additional reviewers.

Reviewer 1 ·

Basic reporting

Thanks to the authors, I think the article is at publication level!

Experimental design

Thanks to the authors, I think the article is at publication level!

Validity of the findings

Thanks to the authors, I think the article is at publication level!

Additional comments

Thanks to the authors, I think the article is at publication level!

Reviewer 2 ·

Basic reporting

None

Experimental design

None

Validity of the findings

None

Additional comments

None

Reviewer 3 ·

Basic reporting

No more questions.

Experimental design

No more questions.

Validity of the findings

No more questions.

Additional comments

No more questions.

Reviewer 4 ·

Basic reporting

Overall though, the writing is clear and technically accurate. Key details like methods, results and references are appropriately reported. The manuscript largely meets PeerJ standards for basic reporting.

Experimental design

The research question is clearly defined and addressing an important problem in early tongue cancer management. The retrospective study design was appropriate given the aim of developing a predictive model. However, more details could be provided on patient selection criteria and how cases were allocated to training and test sets. It is not clear if randomization was truly random. The sample size of 125 patients, with 100 in the training set and 25 in the test set, is relatively small. Larger cohorts would help validate the generalizability of the findings. We'd like to understand how authors decide the number of patients selected for the experiment. Besides, more information on the CT scanning protocol (variables like scanner, slice thickness, contrast use etc.) could provide important context for reproducing the image analysis.

While most methods are described in sufficient detail, more specifics on the deep learning architecture(s) tested, hyperparameters tuned, and model training procedure would strengthen the Experimental Design section. Also, exploring potential biases in the retrospective data that could affect validity is lacking.

In summary, while the goal and general approach are appropriate, more details on study design, datasets, technical methods and performance evaluation could improve transparency and strengthen the experimental standards.

Validity of the findings

The conclusions are well-aligned with the aim of developing a predictive model for OLNM in early tongue cancer. However, they could be strengthened by acknowledging limitations like the retrospective design and small sample size, explicitly stating whether/how the findings constitute meaningful replication or extension of prior work, and clearly linking back to the original research question.The manuscript can also benefit from explicitly assessing the impact or novelty of the findings. I'd say adding a statement comparing the presented multi-omic model to existing approaches would strengthen the evaluation.

Authors did a god job sharing the data to public. The underlying raw CT image and patient data used to develop and validate the models is publicly shared, allowing for independent reproducibility. This open data approach supports robust evaluation.

In summary, while conclusions are appropriately circumspect and limited to results, explicitly addressing impact/novelty and opportunities for meaningful replication would strengthen the validity evaluation per journal standards.

Reviewer 5 ·

Basic reporting

The manuscript is sufficiently well written but there is need to introduce a spacing prior to every bracket that encloses an intext citation in the entire document.

The results projected align to the hypothesis being tested.

Experimental design

The manuscript had well structured research aims to effectively inform important knowledge gaps in the management of tongue cancer.

However, it would have been ideal to incorporate the predictive potential of other diagnostic tests like the molecular testing of OLNM biomarkers at a gene or protein expression level in the clinical model in order to increase the validity of the projected model.


The authors relied on the tumor size and stage where only 2 stages are projected, having other diagnostic indicators of OLNM in comparison to the CT-integrated model would have provided a better rationale for the decision to use surgical operation.

Validity of the findings

• The manuscript has provided a fair statistical evaluation of the results
However, using additional biomarkers of OLNM to correlate to the tumor size and stage would have been superior to comparing professional skills and expertise among the doctors. Years of working experience might not always reflect accuracy or validity of results in clinical diagnosis.
• If 41 out of 125 patients showed OLNM upon post operational pathological examination; What does this say about the decision balance between the urgency to surgically operate and the risk of OLNM in this set of patients?

The CT integrated model and the entire data presentation in the manuscript does not provide any measure on the risk of OLNM for other 84 patients that had been subjected to surgery; It would be nice for the model to be improved further so that it can provide scales of measure for OLNM needed to rule out the need to surgically operate and avoid the associated complications after surgery.

As such it is not clear if the other 84 patients that did not show OLNM in post operation pathological examination in this regard required the surgery that had already been done on them or not.

Additional comments

In general, the study details interesting findings in the pathogenesis and management of tongue cancer.

---

## Round 0.3 · accepted · Accept

All reviewers' comments have been well addressed. It could be accepted.

Reviewer 4 ·

Basic reporting

None

Experimental design

None

Validity of the findings

None

Additional comments

None

Reviewer 5 ·

Basic reporting

The authors have adequately addressed the concerned that i noted during .

Experimental design

All concerns previously noted have been addressed

Validity of the findings

The authors have provided a fair justification for all the projected findings.

Additional comments

I am satisfied with the revisions.